# Pervasive function and evidence for selection across standing genetic variation in *S. cerevisiae*

Christopher M. Jakobson[1], Richard She[1] & Daniel F. Jarosz [1,2]

Quantitative genetics aims to map genotype to phenotype, often with the goal of understanding how organisms evolved. However, it remains unclear whether the genetic variants identified are exemplary of evolution. Here we analyzed progeny of two wild *Saccharomyces cerevisiae* isolates to identify 195 loci underlying complex metabolic traits, resolving 107 to single polymorphisms with diverse molecular mechanisms. More than 20% of causal variants exhibited patterns of emergence inconsistent with neutrality. Moreover, contrary to drift-centric expectation, variation in diverse wild yeast isolates broadly exhibited this property: over 30% of shared natural variants exhibited phylogenetic signatures suggesting that they are not neutral. This pattern is likely attributable to both homoplasy and balancing selection on ancestral polymorphism. Variants that emerged repeatedly were more likely to have done so in isolates from the same ecological niche. Our results underscore the power of super-resolution mapping of ecologically relevant traits in understanding adaptation and evolution.

[1] Department of Chemical & Systems Biology, Stanford University School of Medicine, Stanford, CA 94305, USA. [2] Department of Developmental Biology, Stanford University School of Medicine, Stanford, CA 94305, USA. Correspondence and requests for materials should be addressed to D.F.J. (email: jarosz@stanford.edu)

Since its inception, the central goal of quantitative genetics has been to determine, with ever-greater precision, which loci are responsible for changes to biological traits[1]. This effort has been greatly facilitated by recent advances in genomic sequencing technologies, which have provided genotypes for an unprecedented number of individuals. Ensuing studies have revealed genetic loci associated with a wide array of phenotypes, from height[2] to diabetes risk[3] and even educational outcomes[4]. Although this torrent of data has improved the resolution of quantitative trait locus (QTL) determination, most such associations between genotype and phenotype remain resistant to mapping at the single-gene, let alone single-nucleotide, level without exhaustive experimental follow-up[5–7].

We recently combined experiment and theory to develop a new genetic mapping platform that permits the systematic identification of QTLs at single-nucleotide resolution (*q*uantitative *t*rait *n*ucleotides; QTNs) in the model eukaryote *Saccharomyces cerevisiae*[8]. This effort was initially focused on identifying QTNs responsible for resistance to drugs and other chemical stressors. However, it remains controversial whether such variants are representative of those that drive evolution in the wild because many of the chemical insults in question are unlikely to be encountered by yeast in their natural environments[9].

To address this question, we identify QTNs for genetically complex and ecologically relevant traits in a cross between two wild parental isolates, focusing on metabolic traits critical for the adaptation of *S. cerevisiae* strains to their diverse ecological niches[10–12]. The results of this analysis reveal 107 individual polymorphisms linked to phenotypic diversification, encompassing multiple molecular mechanisms. The complexity of metabolic traits is reflected by the large number of variants of small effect.

Remarkably, and contrary to the expectation that natural variation is dominated by neutral mutations fixed by drift[13,14], we find that many segregating variants are QTNs associated with metabolic or other quantitative traits, and several have patterns of prevalence across the *S. cerevisiae* phylogeny inconsistent with a single emergence event. Extending this analysis to all known variants segregating in *S. cerevisiae*, we find that 53% of the known genetic variants[15,16] are shared by multiple wild *S. cerevisiae* strains. Of these, approximately one-third can be inferred to have emerged multiple times, likely as a result of both homoplasy and balancing selection. Our data therefore suggest that the standing variation in *S. cerevisiae* bears an impress of selection and can be probed to understand the molecular underpinnings of evolutionary change.

## Results

**Mapping QTN that drive complex metabolic innovation**. *S. cerevisiae* strains have recently adapted to a wide range of ecological niches, having shared an ancestor with *S. paradoxus* only some 5–10 million years ago[16–19]. Of the traits thought to be responsible for such adaptation, for instance to anaerobic fermentation or to the host mucosa, one of the most important[10–12] is metabolic innovation. To dissect the genetic basis of growth on diverse carbon sources, we examined a highly inbred cross between parents derived from two different ecological niches: RM11-1a, from a vineyard in California[20], and YJM975, from an immunocompromised patient in Italy[21]. The two strains differ by only 12,054 polymorphisms despite their distinct niches. We previously sequenced the genomes of 1125 $F_6$ haploid progeny of this cross, enabling high-resolution genetic mapping with single-gene, and often single-nucleotide, resolution[8]. We examined the growth of these progeny in quadruplicate on a diverse set of carbohydrates that included sugars and nonfermentable carbon

sources: glucose, galactose, raffinose, maltose, glycerol, ethanol, and sucrose.

We next determined the QTLs responsible for growth on each carbon source. Because complex traits are driven by many alleles of small effect, it can be challenging to identify the underlying QTLs using classical approaches[22]. To map the causal variants for each trait, we used a forward selection procedure, followed by fine mapping using in silico reciprocal hemizygote analysis [Fig. 1a][8]. Of 195 causal loci identified, we could resolve 62.1% ($N = 121$) to single genes (that is, within 1 kbp) and 54.9% ($N = 107$) to single nucleotides [Supplementary Figure 1]. The QTNs responsible for these diverse traits included missense and regulatory variants, but also many synonymous variants in coding regions [Fig. 1b].

The statistical power of our approach enabled us to readily identify causal polymorphisms explaining as little as 0.3% of phenotypic variance [Fig. 1c, Supplementary Figure 1]. Furthermore, because we could identify many QTLs for each trait, we could explain up to 72% of the total phenotypic variance despite the genetic complexity of the traits we examined [Supplementary Figure 1]. Metabolic phenotypes were comparably, and more consistently, complex as compared to growth in the presence of a battery of drugs and other chemical insults (27.7 ± 5.65 QTLs for metabolic traits; 26.0 ± 22.0 QTLs for other traits; mean ± s.e.m.; $p < 0.003$ by *F*-test)[8], emphasizing the genetic complexity of metabolic innovation [Fig. 1c]. Shown in Fig. 1d are example QTNs with diverse molecular mechanisms across the quantitative traits we interrogated.

**Multiple QTNs in a compound QTL for metabolic innovation**. Nonlinearities in the fitness effects of multiple alleles have been known for decades[23], and the predictive accuracy of QTL mapping procedures in yeast can be improved by accounting for these interactions[24,25]. Yet the molecular underpinnings of such nonlinearities often remain mysterious due to the limited resolving power of genotype-to-phenotype mapping. Indeed, most approaches disregard epistatic effects. We sought to harness the power of our technology to characterize these types of interactions. While we lacked the power to survey all QTN–QTN pairs for interacting effects, we did examine our mapping data for instances of QTNs for the same trait residing very close to one another. We reasoned that such cases were likely to represent compound QTLs at which other mapping approaches would likely only resolve a single causal locus.

We noted a striking example of such a locus in a pair of QTNs for fitness in sucrose and raffinose, both located in the *SUC2* gene encoding invertase, a key sucrose metabolic enzyme in *S. cerevisiae*. We expected a priori that one of the QTNs, a frameshift at residue 131 that explained up to 15.6% of the phenotype variance, would impact sucrose metabolism as it is located immediately 5′ of the known catalytic site of Suc2[26]. The pronounced effect of the other QTN, a T > C transition at position −6 in the 5′ UTR that explained up to 12.8% of the phenotypic variance, on the other hand, was more surprising [Fig. 2a]. In concordance with the apparently highly deleterious effect of each polymorphism for sugar metabolism, both were rare across a very large collection of more than 1000 sequenced *S. cerevisiae* isolates [Fig. 2b].

To confirm that both QTNs impinged on Suc2 activity and to link this activity to the observed phenotype, we selected representative segregants to examine in detail. To avoid confounding effects from other segregating QTLs, these were chosen to be isogenic at the other major QTL for growth in raffinose, located at the *ATG19* gene. We first confirmed the growth phenotypes of the segregants on glucose, raffinose, and sucrose: while there was no evident growth defect on glucose,

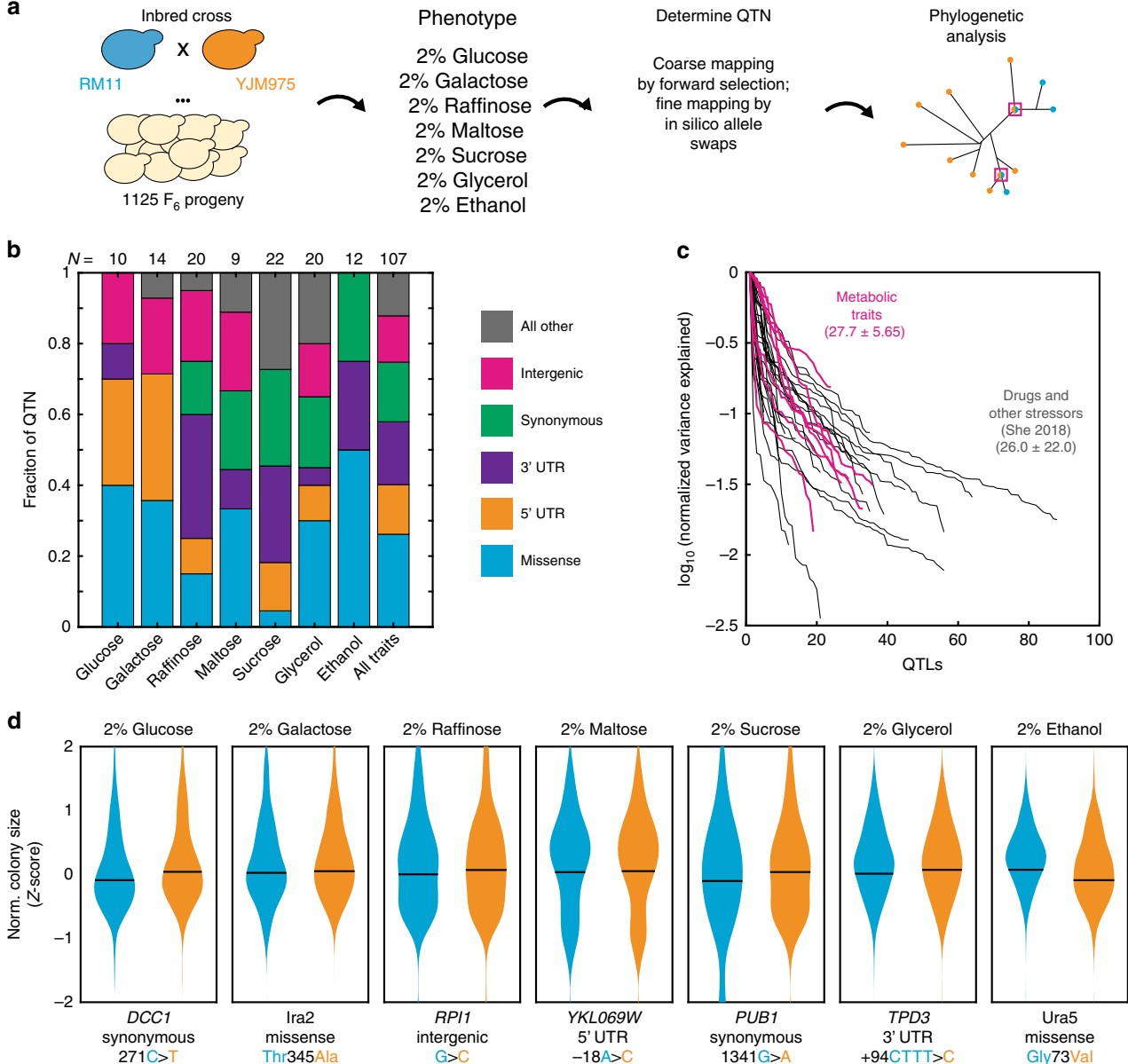

**Fig. 1** Mapping complex metabolic traits to single-nucleotide resolution. **a** Schema of the crossing strategy, phenotyping conditions, genetic mapping, and phylogenetic analysis procedure employed herein. **b** Fraction of QTN of each functional class for each growth condition tested; "all other" includes all other types of polymorphisms, e.g., premature stop codons, frameshifts, loss of a start codon, etc. Indicated at top is the number of QTNs identified for each trait. **c** Variance explained per QTL normalized to the maximum variance explained for each growth condition tested (pink); analogous data for growth in the presence of various drug and other stressors are included as a reference for the complexity of drug-resistance traits[8] (gray). Also indicated is the mean ± s. e.m. of number of QTLs identified per trait for each class. **d** Effect of selected example QTNs in conditions as indicated. Shown are normalized, Z-scored colony size for each allele. Line indicates the mean. Blue: RM11 allele; orange: YJM975 allele. Source data are provided as a Source Data file

both QTNs were associated profound growth defects on both sucrose and raffinose [Fig. 2c]. This is consistent with a defect in sucrose catabolism, as the trisaccharide raffinose is first decomposed into galactose and sucrose. To further confirm that Suc2 activity was the molecular phenotype responsible for the growth defect, we assayed the total cellular invertase activity in each of the segregants when propagated in raffinose and sucrose. In concordance with our hypothesis, segregants with either or both QTNs exhibited greatly reduced invertase activity [Fig. 2d]. Indeed, segregants bearing the nonparental ditypes at the two loci were comparably compromised as compared to the YJM975 ditype. While we did not exhaustively survey the genetic variation in our mapping panel for such interactions, our finding of a

strong compound QTL in our limited search suggests that the phenomenon may be common. Continued improvements in the resolution of genetic mapping approaches will likely reveal many more examples of compound QTLs, as were recently described for genetic variation segregating in the BY4741 and RM11 strains[27].

**Evidence of positive selection on metabolic traits**. Although the traits we examined are likely to be important in many environments, it is impossible to completely recapture in the laboratory the selective forces that have driven *S. cerevisiae* evolution in the wild. We therefore turned to a powerful statistical test for

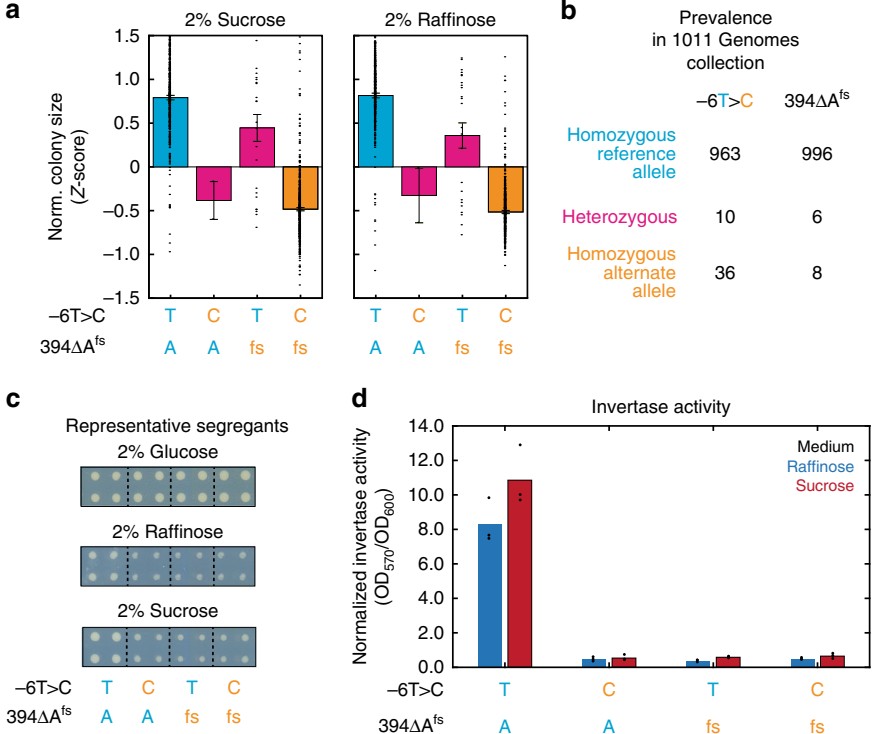

**Fig. 2** Neighboring QTNs in a compound QTL. **a**) Growth phenotypes of segregants with the parental ditypes (*SUC2*[−6T/394A], blue; *SUC2*[−6C/394fs], orange) and nonparental ditypes (*SUC2*[−6T/394fs] and *SUC2*[−6T/394fs], pink). Data shown are mean ± s.e.m of normalized, *Z*-scored colony size for each ditype. **b** Prevalence of the causal *SUC2* variants shown in (**a**) across 1011 sequenced *S. cerevisiae* isolates[28]. **c** Growth of representative segregants with genotypes as in (**a**). Shown are technical quadruplicates arrayed in squares; panels are representative of *N* = 8 biological replicates. **d** Normalized invertase activity of representative segregants with genotypes as in (**a**) when grown in media containing raffinose (blue) or sucrose (red), as indicated. Data shown are *N* = 3 biological replicates; bars show the mean. Source data are provided as a Source Data file

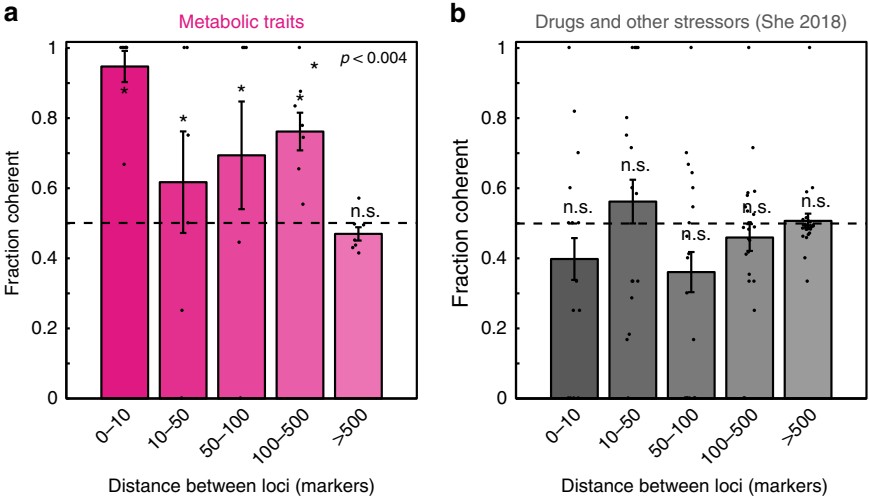

**Fig. 3** Directional test for selection on metabolic and other traits. Fraction of alleles from the same parent with coherent effects on phenotype as a function of distance between QTLs (in markers) for **a** metabolic traits examined here and **b** drugs and other stressors examined previously. Shown is mean ± s.e.m. across all mapped traits. *p* values by binomial test against random expectation (shown in black dashed line). Source data are provided as a Source Data file

selection, inspired by Orr and predicated on the idea that positive selection on a given trait in one lineage should enrich it for QTLs of coherent effect[28,29]. This test is implemented by calculating the fraction of variants from the same parent that have a coherent effect as a function of genetic distance and comparing this enrichment to the random expectation of no coherence. We observed a striking enrichment for nearby variants from the same

parent to have the same effect on phenotype (*p* < 0.004 by binomial test) even at distances of up to 500 markers [Fig. 3a]. Moreover, the same was not true of QTLs for the battery of drugs and other stressors tested previously [Fig. 3b]. While we previously found that such QTLs are closer to one another than expected by chance (possibly reflecting a ghost of ancient operons[8]), they have evidently not been subject to sufficient selective

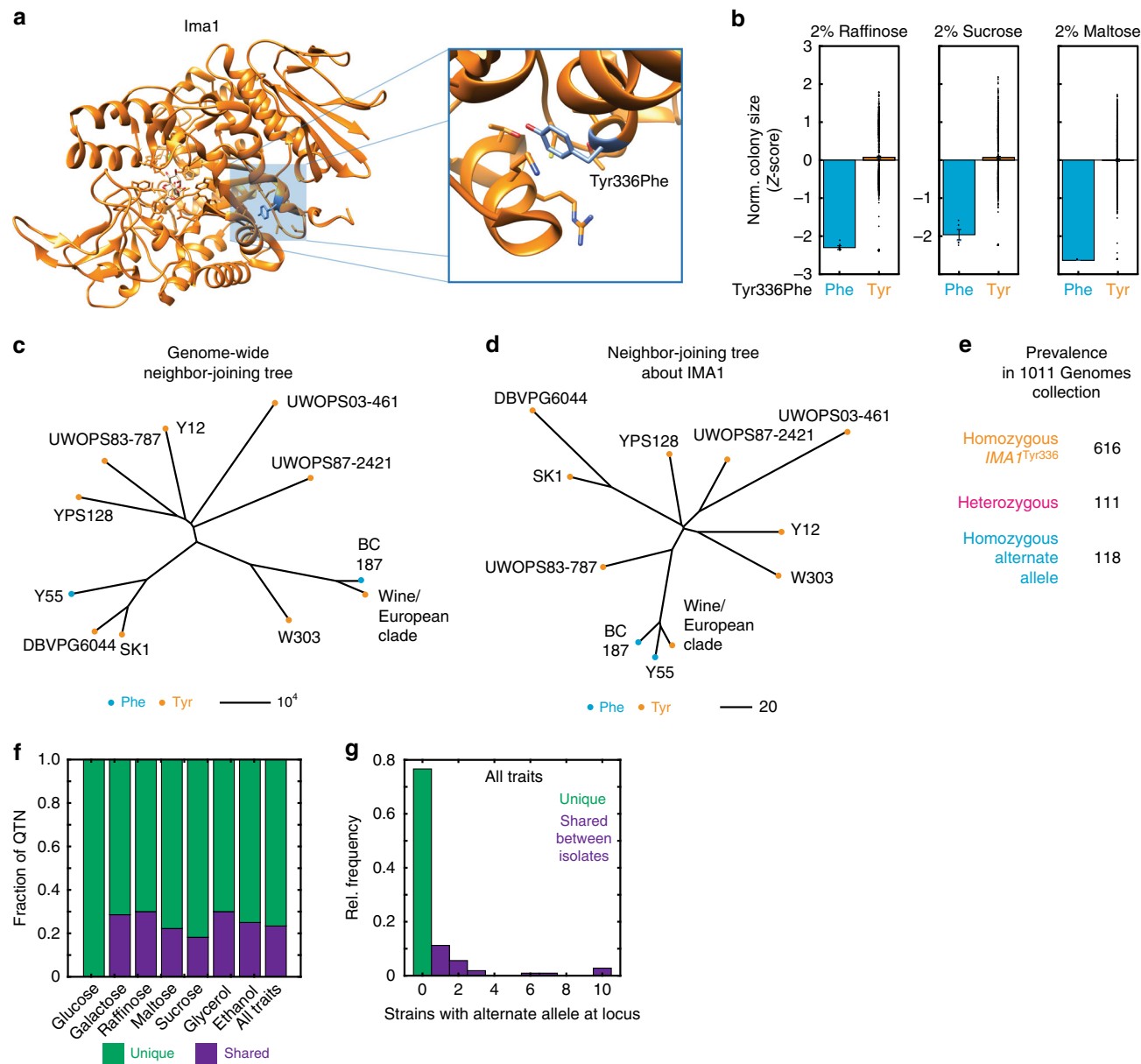

**Fig. 4** Prevalence of *IMA1*^Tyr336Phe and other QTNs across *S. cerevisiae*. **a** Crystal structure of Ima1 in complex with isomaltose; highlighted is the Tyr336 residue that is mutated to Phe (PDB ID: 3AXH). **b** Growth phenotypes of segregants with Phe336 (blue) and Tyr336 (orange) in raffinose, sucrose, and maltose. Data shown are normalized, Z-scored colony size for each allele; bars show the mean. **c** Phylogeny of the *IMA1*^Tyr336Phe variant across the SGRP collection; the neighbor-joining tree is constructed on the basis of all segregating polymorphisms. **d** Phylogeny of the *IMA1*^Tyr336Phe variant across the SGRP collection; the neighbor-joining tree is constructed on the basis of only the segregating polymorphisms within 250 variants of the *IMA1* variant on the genome. Scale bars show neighbor-joining distance. **e** Prevalence of the *IMA1* variant shown in (**a**) across 1011 sequenced *S. cerevisiae* isolates[28]. **f** Fraction of QTN that are unique to RM11 or YJM975 (singletons; green) or shared with another SGRP isolate (purple), for each growth condition tested. **g** Histogram of the number of strains within the SGRP collection bearing the alternate allele at each locus identified as a QTN. Source data are provided as a Source Data file

pressure to result in detectable coherence. Together these data suggesting that the metabolic phenotypes mapped here are on average more ecologically relevant to *S. cerevisiae* in the wild.

**Many QTNs are not unique to the parent strains**. While directional tests can address the question of whether traits have been subject to positive selection, it remains unclear which molecular variants in particular may have been subject to enrichment. Phylogenetic analysis, aided by recent growth in the availability of wild yeast genome sequence data[15,16,18], can thus

provide powerful complementary information. To assess the ecological relevance of our causal variants, we examined the prevalence of our QTNs in a phylogenetically diverse panel of wild yeast isolates from many different ecological niches[15,16]. As one might expect, some QTNs were unique to RM11 or YJM975. By contrast, 25 QTNs were not singletons, but were instead present in multiple *S. cerevisiae* strain backgrounds. For example, the Tyr336Phe missense mutation in Ima1, an enzyme of the isomaltase family[30], conferred improved growth on raffinose and sucrose. *IMA1* is important for growth on sucrose in the absence of *SUC2*, and here may be playing a similar "moonlighting"

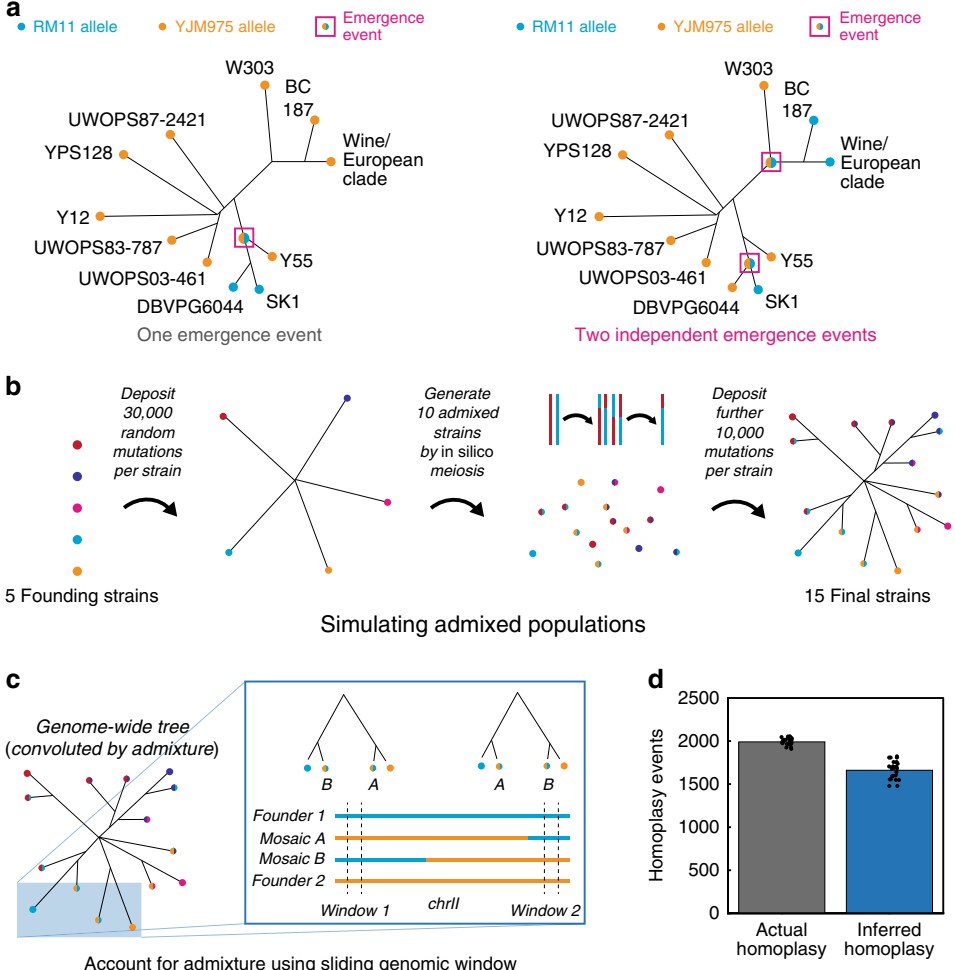

**Fig. 5** Using local phylogeny to accurately infer homoplasy. **a** Example phylogeny of a hypothetical variant that (left) emerged only once and (right) independently emerged twice. **b** Schematic of the in silico mutagenesis and mating procedure used to generate simulated admixed populations. **c** Schematic of the sliding phylogenetic window approach used to account for admixture. **d** Actual and inferred homoplasy based on this approach for $N = 25$ simulated populations with structure similar to that of the SGRP collection; bars show the mean. Source data are provided as a Source Data file

role[31]. The mutation, located distal to the enzyme active site [Fig. 4a], may disrupt the packing of the Tyr residue, which ordinarily is oriented toward the interior of the enzyme with its hydroxyl group in proximity to the hydroxyl of Thr290[32]. The Phe variant, despite being highly deleterious to growth on raffinose, sucrose, and maltose [Fig. 4b], is present in several isolates from the *Saccharomyces* Genome Resequencing Project (SGRP)[15] [Fig. 4c]. Also shown in Fig. 4d is a neighbor-joining tree constructed for the genomic neighborhood of *IMA1*, showing that the Tyr336 variant appears to have re-emerged in the wine/European clade even after accounting for mosaicism among SGRP isolates. This variant also proved to be common across a larger collection of more than 1000 isolates[33], with more than 200 strains bearing at least one copy of an alternate allele to the reference Tyr336 variant [Fig. 4e]. Extending this analysis to causal variants for all traits examined here, 23.3% of metabolic the QTNs were shared by other isolates in the SGRP collection [Fig. 4f], and were sometimes present in multiple strains [Fig. 4g].

**Phylogenetic evidence for selection on QTNs**. The apparent re-emergence of the *IMA1*^Tyr336Phe allele raised the intriguing possibility that independent emergence events have repeatedly produced the same genetic innovations across the *S. cerevisiae* phylogeny. To test this hypothesis, we examined all the metabolic

innovation QTNs for evidence of selection[34]. We have reported above and previously[8] that many QTNs are synonymous variants. Accordingly, we avoided methods that assume the neutrality of synonymous mutations (e.g., $K_a/K_s$ and McDonald–Kreitman criteria[35]). Instead, we exploited the fact that detailed genomic and phylogenetic information is available for many wild yeast isolates, allowing the inference of repeated emergence by direct comparison of variant and strain phylogeny[15,16]. The concept of our analysis is simple: if the distribution of alleles on the strain tree is inconsistent with an allele emerging on only one branch, we can in principle infer that the allele has been subject to positive selection leading to multiple fixation events [Fig. 5a]. However, there are several key complications that must be accounted for.

First, it is known that the sequenced isolates of the wild strain collection have been subject to substantial admixture, with many strains evidently being the product of mating between other sequenced isolates. Thus, the whole-genome neighbor-joining phylogeny is not consistent with the phylogeny based on particular chromosomes or chromosomal regions. This discrepancy will lead to spurious apparent multiple emergence if not accounted for. We therefore adopted a sliding-window approach to determining phylogeny, building local neighbor-joining trees in the vicinity of each segregating polymorphism based on a 500 variant-wide sliding window. To confirm that this approach

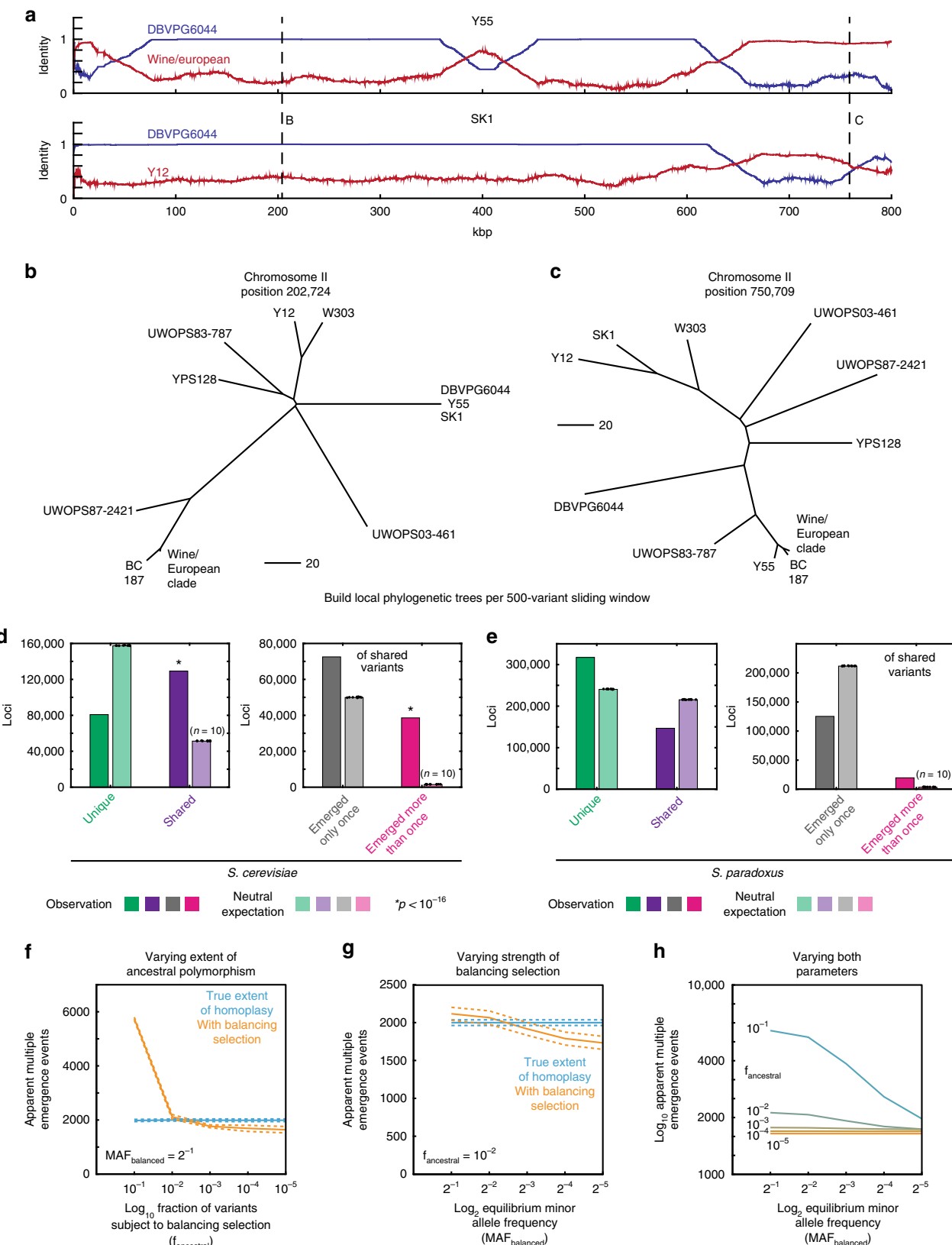

Build local phylogenetic trees per 500-variant sliding window

properly accounts for the possibility of admixture, we conducted simulations based on in silico populations quantitatively similar to the wild yeast collection. Briefly, each simulation began by instantiating 30,000 random mutations in each of five founder strains [Fig. 5b]. Next, ten mosaic strains were generated by random mating of the five founders, followed by in silico meiosis

consisting of one crossover per chromosome and the generation of haploid progeny, of which one was retained. Finally, a further 10,000 random mutations were deposited in all 15 simulated isolates. We can then assess the true extent of homoplasy and compare it to that detected by our inference procedure. Across $N = 25$ such simulations, we found that our procedure did

**Fig. 6** Widespread apparent multiple emergence of natural variation. **a** Plots of sequence identity (500-variant sliding window) of Y55 (top) and SK1 (bottom) to DBVPG6044 (blue) and wine/European clade modal genotype or Y12, respectively (red). Neighbor-joining trees for all strains in the SGRP collection based on 500-variant sliding windows centered about **b** chrII position 202,724 and **c** chrII position 750,709. Scale bars show neighbor-joining distance. Fraction of variants segregating within all strains of the SGRP **d** *S. cerevisiae* and **e** *S. paradoxus* collections that are (left) unique (green) or shared with another isolate (purple) and (right) inferred by phylogeny to have emerged only once (gray) or more than once (pink). The neutral expectation in the absence of selection was calculated separately for each strain collection by ten independent simulations; the bars show the mean across ten simulations and the results of each of the ten simulations are shown; $^{*}p < 10^{-16}$ by permutation test. **f** True extent of homoplasy (blue) as compared to inferred apparent multiple emergence (orange) as a function of the fraction of polymorphisms that are subject to balancing selection ($f_{ancestral}$). $MAF_{balanced}$ is set to $2^{-1}$. **g** True extent of homoplasy (blue) as compared to inferred apparent multiple emergence (orange) as a function of the strength of balancing selection ($MAF_{balanced}$). $f_{ancestral}$ is set to $10^{-2}$. **h** Inferred apparent multiple emergence as a function of both $f_{ancestral}$ (as indicated; shaded lines) and $MAF_{balanced}$ (abscissa). Results shown are mean ± s.e.m. Source data are provided as a Source Data file

account for admixture, and indeed returned a mild underestimate of the extent of homoplasy (since a fraction of homoplasy events are phylogenetically indistinguishable from a single emergence) [Fig. 5c, d]. Moreover, our approach accurately captured known mosaicism in the wild strain collection, e.g., that of chromosome II of the Y55 and SK1 strains[15,16,33] [Figure 6a–c].

We therefore applied our approach to infer the extent of apparent multiple emergence of the QTNs we identified. As noted above, 23.3% of QTNs from each condition were shared with other SGRP isolates [Fig. 4f, g]. When we included QTNs identified previously[8] we found that 26.8% ($N = 128$) of variants identified as QTNs in our cross are shared with other yeast isolates. Of these, 21.1% ($N = 27$) of these apparently emerged multiple times in *S. cerevisiae*.

**A signature of widespread selection across natural variation.** Drift-centric standard models of population genetics, while intuitively appealing and analytically tractable, are often at odds with the rapid phenotypic diversification typical of microbes like *S. cerevisiae*[13,36]. The findings described above led us to wonder whether a larger fraction of genetic variation in *S. cerevisiae* has been subject to selection than is generally thought[37]. Therefore, we analyzed all variants segregating in the wild strain collection using our emergence inference method, in which the phylogeny was recalculated for each variant based on a sliding genomic window [Figure 6b, c]. In total, 112,285 of 210,363 variants (53.5%) were shared, of which 38,549 (34.3%) were inferred to have emerged more than once [Fig. 6d]. The neutral expectation in the absence of selection predicts only $51,334 \pm 201$ shared variants and $1452 \pm 30$ parallel emergence events (mean ± s.d. for $N = 10$ simulations). This strongly suggests that a substantial amount of the extant variation in *S. cerevisiae* has been subject to selection.

To confirm that the pattern of variant sharing we observed was not a sampling artifact of the strain collection we examined, we also assessed the sharing of variants in the 1002 Yeast Genomes Project collection[33]. In concordance with the observations described above, only 44% ($N = 772,398$) of variants were singletons, whereas 56% ($N = 982,468$ loci with homozygous alternate allele) occurred in at least two isolates [Supplementary Figure 2]. This observation is striking in light of the declining numbers of distinct variants observed with increasing numbers of sequenced isolates [Supplementary Figure 2] and is consistent with strong purifying selection having acted on abundant variation in the *S. cerevisiae* lineage.

Interestingly, the pattern of apparent multiple emergence was quite different in the nearest relative of *S. cerevisiae*, *Saccharomyces paradoxus*, for which genome sequences of a 23-strain *S. paradoxus* collection are available[15]. In contrast to the *S. cerevisiae* strain collection, whose members are ecologically diverse, all but one of the sequenced *S. paradoxus* strains were isolated from *Quercus* (oak trees). Others have suggested that this

may have resulted in reduced selective pressure among the *S. paradoxus* isolates to adapt to new ecological niches[38], and we reasoned that this might in turn have reduced the parallel emergence of novel adaptive variants. We analyzed the prevalence of multiple emergence events in this collection (which harbors 464,307 segregating variants) by the second inference method described above and again determined phylogeny locally for each variant using a 200-variant sliding window. In contrast to our findings for *S. cerevisiae*, only 146,819 variants were shared between isolates, and 19,628 multiple emergence events were inferred [Fig. 6e]. The neutral expectation was of $215,300 \pm 354$ shared variants and $3429 \pm 43$ emergence events (mean ± s.d. for $N = 10$ simulations). The relative paucity of apparent multiple emergence events in *S. paradoxus*, the closest extant relative of *S. cerevisiae*, suggests that ecological diversification has played a role in generating the parallelism that we found to be commonplace in *S. cerevisiae*.

**Balancing selection as an alternative to multiple emergence.** Homoplasy is not the only explanation for the prevalence of apparent multiple emergence events. The last common ancestor of the wild isolates we analyzed likely harbored substantial polymorphism, presenting two other important sources of modern genetic variation: neutral polymorphisms maintained due to incomplete lineage sorting[39] and functional polymorphisms maintained by balancing selection[40]. The first possibility is likely not a major contributor, as most neutral variation is expected to resolve to reciprocal monophyly within 10 $N_e$ generations (only ~100,000 years for *S. cerevisiae*)[39,41]. The latter, on the other hand, is important to consider as an alternative to homoplasy. We therefore simulated this possibility for a range of strengths and extents of balancing selection. Following the same procedure as above, we first constructed highly mosaic in silico yeast populations consisting of 15 strains and $N_{total}$ ~300,000 total random mutations population-wide. In the context of these strains, we simulated the addition of $f_{ancestral}N_{total}$ ancestral polymorphisms maintained at a balancing minor allele frequency of $MAF_{balanced}$. For each polymorphic locus, the "sequenced" locus detected in each strain was simulated by a random draw based on $MAF_{balanced}$.

Little is known regarding the quantitative strength and extent of balancing selection in the wild[40], so we simulated the presence of balanced polymorphisms using parameter estimates spanning several orders of magnitude ($f_{ancestral} = 0.1–10^{-5}$; $MAF_{balanced} = 0.5–2^{-5}$). As expected, increasing both the fraction of loci subject to balancing selection and the equilibrium minor allele frequency increased the extent of apparent multiple emergence inferred from the simulated genotype data [Fig. 6f–h]. Quantitatively, however, even for substantial balancing selection ($f_{ancestral} = 0.1$ and $MAF_{balanced} = 0.5$) the simulated extent of apparent multiple emergence was less than that observed in the wild-strain collection: only $5743 \pm 57$ (mean ± s.d.; $N = 5$ simulations) apparent multiple

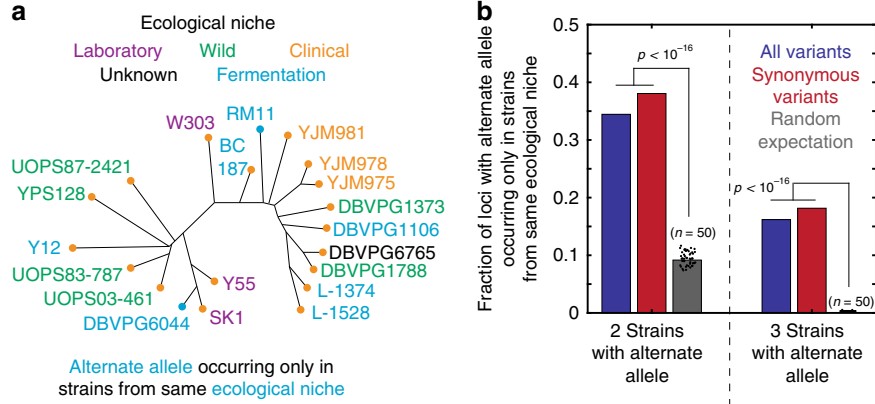

**Fig. 7** Enrichment of shared alternate alleles in common ecological niches. **a** Schematic of a phylogeny in which the alternate (RM11) allele occurred twice, and only in the fermentation niche. **b** Fraction of variants segregating in the SGRP collection for which the alternate allele occurs only in strains from same ecological niche (blue: all variants; red: only synonymous variants), for loci with the alternate allele occurring (left) in two strains or (right) in three strains; the neutral expectation was calculated by 50 independent simulations; the mean of the simulations and the results of each simulation are shown in gray and with black dots, respectively. p values by permutation test. Source data are provided as a Source Data file

emergence events were inferred. This suggests that homoplasy is likely a key contributor to the observed level of apparent multiple emergence (38,549 apparent events), even in the presence of high levels of ancestral polymorphism and balancing selection.

On evolutionary timescales, the likelihood of a given nucleotide mutation having occurred at some point in a *S. cerevisiae* population is high. Each base pair should be mutated approximately every $10^3$–$10^4$ generations in a typical *S. cerevisiae* effective population size ($N_e$ ~$10^6$) with a mutation rate of $5 \times 10^{-10}$ per base pair per division[42,43]. This notion is supported by the reproducible advent of beneficial genetic variants in laboratory evolution experiments[44–47]. On evolutionary timescales, therefore, the exploration of the genotypic space near the reference genome is likely nearly complete. However, the wild strains we examined contain only ~240,000 segregating variants across a genome of approximately 12 Mb (~2% of loci) and RM11 × YJM975 contains only ~12,000 variants (~0.1% of loci). Even the very large 1002 Yeast Genomes Project collection contains only ~1,700,000 total variants (~14% of loci), an even smaller fraction relative to the total number of strains sampled [Supplementary Figure 2][33]. These patterns suggest that *S. cerevisiae* has been subject to strong purifying selection[15]. Our conservative analysis, based on the inference of apparent multiple emergence events, concludes that a substantial number (>5%; $N = 27$) of all QTN we have identified thus far (in a limited survey of growth conditions) have been subject to selection in natural *S. cerevisiae* populations. This estimate is a conservative lower bound on the fraction of variants in our cross that have been subject to selection, but still suggests RM11 × YJM975 alone harbors at least 600 ecologically relevant variants, and likely many more.

**Selection for adaptation to ecological niche.** Finally, we investigated connections between the phylogenetic evidence for selection and the actual selective forces at play in nature. If adaptation to ecological niche is a relevant selective pressure, variants adaptive in a certain niche should be enriched in isolates from that environment. Therefore, we evaluated the enrichment of alternate alleles in strains isolated from particular ecological niches. For every shared variant, we assessed whether the multiple occurrences were more likely to have occurred in strains isolated from the same ecological environment ($N = 26,671$ and $N = 20,089$ variants present in two and three strains, respectively)

[Fig. 7a]. Indeed, both doubly and triply occurring alternate alleles were far more likely to have occurred solely within a single niche than would be expected by chance ($p < 10^{-16}$ by permutation test) [Fig. 7b]. The enrichment was slightly stronger when only synonymous variants were considered ($N = 10,014$ doubly occurring and $N = 7599$ triply occurring variants). Taken together, these data provide strong evidence that a substantial subset of standing variants in *S. cerevisiae* have been subject to selection (and have presumably been beneficial), facilitating the recent adaptation of this organism to diverse environments.

## Discussion
Using a highly inbred yeast cross consisting of 1125 $F_6$ progeny of known genotype, we identified 107 QTNs for seven ecologically relevant and genetically complex metabolic traits. These QTNs are of multiple variant types and exhibit complex interactions with one another, including nonlinearities within a typical QTL resolution window. A total of 25 of the metabolic QTNs are present in multiple wild strains, and, surprisingly, many of these can be inferred from the phylogeny of sequenced isolates to have emerged multiple times. Together, these findings strongly suggest that these variants have been subject to selection, resulting in either homoplasy or balancing selection on ancestral polymorphism.

As recently as 2013, only about 100 quantitative trait genes, and a small handful of QTNs, had been identified in *S. cerevisiae*[48]. This is in contrast to deletion and overexpression studies, which have frequently detected hundreds of genes that affect a given trait[49]. Here, we demonstrate that many QTNs, often of small effect, can readily be detected for highly complex metabolic traits. Even those loci not unambiguously resolved to a single QTN can, in most cases, be resolved to a window spanning a single open reading frame (ORF) and associated regulatory sequences, allowing the systematic and mechanistic determination of the particular molecular variation responsible for changes in phenotype.

Many of the QTNs we discovered would likely be disregarded based on the conventional wisdom regarding the effects of mutations. For instance, more than 15% of QTNs are synonymous variants within ORFs, which are assigned low probabilities of affecting phenotype by algorithms such as SIFT and PolyPhen-2[50,51] but likely affect mRNA or protein levels in certain cases. Although variants predicted to be consequential likely do have outsize (often negative) effects on protein function, an approach

that is agnostic to variant type may be appropriate when attempting to predict which of multiple candidate causal variants within a statistical confidence window is the true causal allele.

By improving our statistical resolution to the single-nucleotide scale, we teased apart interactions between variants located within dozens of nucleotides of one another on the yeast chromosome. These interactions may be hidden from traditional approaches with lower resolving power. This phenomenon may partially explain an apparent paradox: epistasis is common in individual molecular examples[52], yet linear QTL models accurately describe phenotype in many cases[53]. Some bona fide nonlinearities may in fact be masked within haplotype blocks[54].

Our approach provides high-resolution understanding of the genotype-to-phenotype relationship in the model microbe *S. cerevisiae*, but a question remains, are the QTNs we discover germane to the process of evolution as it occurs in real environments? This issue is central to extending our findings to multicellular eukaryotes not susceptible to the inbreeding strategy that is key to our approach[8,9]. To address this critical connection from the laboratory to the outside world, we combined our QTN measurements with detailed genomic and phylogenetic knowledge of wild *S. cerevisiae* isolates.

Although some of the QTNs are singletons, more than 25% were shared with other sequenced isolates. We inferred that more than 20% of the shared QTNs we have identified are not only shared with other strains, but have apparently emerged multiple times during the evolutionary history of the phylogenetically diverse isolates. This strongly suggests that these variants have been subject to selection in wild-yeast populations, resulting in apparent multiple emergence either as a result of homoplasy or balancing selection on ancestral polymorphism. Moreover, we observed a similar signature of selection in standing variation across a collection of wild-yeast isolates. The rate at which variants were inferred to have emerged multiple times in this sampling of wild yeast was dramatically higher than would be expected by chance, given the known phylogeny and genome size of the isolates. This could be a result of at least two different kinds of selective pressure: positive selection on the fitness advantages conferred by the multiply emerging variants; and evolutionary constraint resulting from purifying selection against detrimental alternatives at a given nucleotide position. Both modes of selection may play a role in species, such as *S. cerevisiae*, in which exploration of variants in the neighborhood of the reference genome is likely to be extensive. At present, our analysis cannot distinguish between these two forms of selection.

The simulations of complex populations with mutation, balancing selection, and admixture, against which we benchmarked our inference method, are substantial simplifications of the complexity of evolution in the wild. There are no doubt some aspects of the population history of *S. cerevisiae*, and the nature of molecular evolution, that are incompletely accounted for in such analyses. For instance, a small number of regions in the genome may be subject to extraordinary rates of mutation[55,56], and it is likely that some admixture events were incompletely accounted for. However, while the inferred history of any one polymorphism may thus be inaccurate in some details, we nonetheless expect that our overall analysis, and the pronounced trends we observe, are broadly robust to such complications.

Parallelism has recently been observed in various other organisms and viruses, including *Arabidopsis*[57], *Drosophila*[58], herbivorous insects[59], echolocating mammals[60], and influenza[61]. Our results suggest not only that the phenomenon is pervasive in *S. cerevisiae*, but also that it can encompass all types of molecular variation (not just changes to protein coding sequences). Moreover, quantitative estimates of the extent of apparent multiple

emergence suggested that both balancing selection and homoplasy are likely at play.

The combination of apparent parallelism and balancing selection observed in *S. cerevisiae* stands in contrast to the results of an analogous analysis of a collection of *S. paradoxus* strains, which revealed many fewer apparent parallel emergence events despite much greater genetic diversity. This difference may in part be explained by the different ecological histories of the sampled populations: the sequenced *S. cerevisiae* strains are from diverse ecological niches, whereas the *S. paradoxus* strains are not. As others have noted previously[15,38], the disparate ratios of genotypic to phenotypic variation in the two species is inconsistent with the accumulation of phenotypic variation primarily due to the fixation of polymorphisms by drift. Instead, our results suggest that the recent specialization of *S. cerevisiae* to many niches may have driven the parallel emergence of novel adaptive variants across genetically isolated populations. Moreover, this adaptation appears to have been driven by a wide array of molecular mechanisms, including many synonymous variants. Other types of genetic variation, such as structural rearrangements[38], likely also play a role; indeed, these may have been subject to much the same positive or constraining selection.

This finding is in accordance with prior observations that the same adaptive mutations repeatedly emerge in independent populations in laboratory evolution experiments[46,62,63] and the anecdotal observation that standing variation in wild yeast isolates is frequently adaptive[64]. The effective population size of yeast dictates that *S. cerevisiae* (and likely other microbes with relatively large populations and relatively small genomes) readily explore the genotypic space near the ancestral genome[43], and can thereby access new genetic variants that are subsequently subject to substantial selective pressure.

Further exploration of the "super-resolution" genotype-to-phenotype map in diverse strains, species, and genera will be a fruitful avenue to establish the generality of our findings. Two important considerations should be taken into account, however, in generating mapping populations with increasing nucleotide distance between the parents. First, the number of haploid progeny analyzed must increase to maintain sensitivity to variants of small effect. Second, the number of meiotic crossovers must also be increased, to maintain similar levels of linkage disequilibrium and maintain fine-mapping resolution. (A more detailed mathematical description of these tradeoffs can be found in She and Jarosz[8].) Two strategies therefore suggest themselves: creating several similarly sized mapping populations from founding parental pairs on distal branches of the *S. cerevisiae* tree (or that of another fungus), or generating very large, even more inbred crosses that maintain statistical power even in the presence of sufficient polymorphism to span the "gaps" between lineages. The practicality of these approaches will be constrained by strategies for both genotyping and phenotyping.

Our findings paint a hopeful picture for the future of the QTN program in yeast quantitative genetics. Indeed, the strong selection (due to large effective population size)[65], recent ecological divergence[16,66], and genetic tractability of *S. cerevisiae* (and the Saccharomycotina more generally[67]) make budding yeast an excellent model for studying how molecular diversification translates into selectable phenotypic change. These key advantages have two important ramifications. First, much of the genetic variation in *S. cerevisiae* has been subject to selection, and therefore contains a signature of molecular evolution. Second, the phenotypic outcomes of ecologically relevant natural variation can be assigned at the nucleotide level using our technology[8] and other approaches[68]. Continuing studies dissecting this variation, therefore, have the potential to access the molecular genetic and biochemical underpinnings of real evolutionary processes.

## Methods

**Yeast strains, growth conditions, and growth measurements.** Parental genotypes are described in Supplementary Table 3. Genotyped haploids were revived from glycerol stocks by inoculation into 200 μL liquid yeast peptone with dextrose medium (YPD) in 96-well plates, and then grown to saturation for ~48 h at 30 °C without agitation. Cells were resuspended by repeated pipetting and transferred from saturated cultures onto Singer PlusPlates with solid YPD (with 2% agar) in 384-spot format. These were again incubated for ~48 h at 30 °C. Cultures were then replica-pinned to appropriate solid growth medium (carbon sources as indicated, with 2% agar, 6.7 g/L yeast nitrogen base without amino acids and 0.79 g/L CSM nutrient supplement) in biological duplicate using a Singer Rotor robotic pinning instrument. Solid cultures were grown at 30 °C and imaged every 24 h for a total of 4 days. Media composition is described in Supplementary Table 2. Colony sizes were determined from plate images using the SGATools platform[69] and mean colony sizes for each genotype were determined using custom code implemented in MATLAB.

**Measurement of cellular invertase activity.** The invertase assay was carried out according to the manufacturer's directions (Sigma MAK118). Briefly, cells were grown to saturation, washed, and resuspended at $OD_{600}$ ~0.1 in media as indicated. Cells were propagated for 4 h at 30 °C, washed, diluted 1:10 into assay buffer containing sucrose, and incubated for 20 min. At this point, reaction buffer containing the assay enzyme mixture and the chromogenic substrate were added and reacted for 5 min. Invertase activity was calculated as $OD_{750}/OD_{600}$.

**Determination of QTLs and QTNs.** QTLs for each trait were determined using custom code implemented in MATLAB[8]. Briefly, genotypes for all 1125 $F_6$ progeny were determined at 12,054 segregating sites of variation. Pseudo-genotypes were appended to reflect plate number and plate position to account for geometric effects and plate-to-plate variations in inoculum density. Phenotypes (i.e., colony sizes) were calculated as the mean of biological duplicates, then normalized and Z-scored based on the standard deviation before regression. A forward selection scheme with an initial inclusion criterion of $p < 10^{-3}$ (by F-test) was used to select genetic predictors of measured phenotype based on the genotype matrix. Only terms with final $p < 10^{-5}$ (by F-test) were retained after completing the selection procedure. These loci were subsequently fine-mapped using in silico allele swaps to define the QTN score in the vicinity of each QTL. The mathematical details of the analysis of variance (ANOVA) procedure used for fine mapping are described elsewhere[8]. Briefly, the procedure tests against the null hypothesis that the variant in question is not causal (relative to all other nearby segregating variants); example QTN scores for resolved and unresolved loci are shown in Supplementary Figure 1. Loci that could not be unambiguously resolved to a single QTN were omitted from subsequent analysis. False-discovery rates were estimated empirically by regression against randomly permuted real data; false discovery at the QTL level was estimated to be less than 10% for all traits examined [Supplementary Table 1]. A complete table of all discovered QTLs and QTNs can be found as tab-delimited text in Supplementary Data 1.

**Analysis of wild S. cerevisiae isolate genotypes.** The genotypes of the 19 wild isolates of the SGRP[15] were determined with reference to the variant call (.vcf) file available from http://www.moseslab.csb.utoronto.ca/sgrp/download.html. Only the 211,804 loci (of $N = 240,765$ total) with sequence coverage for at least 18 isolates were considered. Genotypes of these strains at loci with variants segregating in the RM × YJM cross were determined based on the chromosome and nucleotide positions of the variants. "Shared" variants are those for which the SGRP collection contains at least one instance of the alternate allele at that locus. Additional alternate alleles beyond those segregating in RM × YJM were not considered in this calculation. Also considered for some analyses were the genotypes of the 1012 S. cerevisiae isolates of the 1011 Yeast Genomes Project collection[28]. Genotypes were calculated in a similar manner from the.vcf file describing the segregating variants within this collection of wild isolates.

**Inference of independent allele emergence events.** The strains of the wine/European clade (YJM981, YJM978, YJM975, DBVPG1373, DBVPG1106, DBVPG6765, DBVPG1788, L-1374, and L-1528) are extremely closely related and subject to mosaicism such that their phylogeny cannot reliably be determined. Therefore, we considered the modal genotype of this clade to be the genotype at the root of the clade with respect to the SGRP tree and did not infer any independent emergence events within the wine/European clade. To control for mosaicism, phylogeny was recalculated for each variant based on a 500-variant sliding window centered on the variant in question. Phylogeny was inferred by the neighbor-joining method using SNP differences[16]. The number of independent emergence events for the remaining shared variants was inferred by a maximum likelihood estimate of the genotype of the last common ancestor at each branch point in the phylogenetic tree: if the two children of a node differed in genotype, then an independent emergence event was assigned to that node [Figure 4a, b]. Otherwise the node was assigned the genotype of the children. In the case when the children of a node comprised an emergence event at one child and a genotype at the other,

the called genotype was assigned to the parent node. Only variants with high-quality sequence coverage for all isolates were considered.

**Generation of random mutations.** To simulate the random expectation of mutation distribution under neutrality, single-nucleotide polymorphisms were generated based on the reference genome of S. cerevisiae strain S288C and a transition-to-transversion ratio of 3. Indels, gross chromosomal rearrangements, and other types of mutations were neglected. The nucleotide to mutate was chosen at random; reversion within individual simulated "isolates" was neglected as it very rarely occurred under the parameter combinations simulated.

**Neutral model of variant distribution in the absence of selection.** To simulate the neutral expectation for the frequency of apparent multiple emergence events within the SGRP collection in the absence of selection, we randomly distributed 240,000 random variants as described above. Variants were then propagated to all extant isolates descended from the node at the end of the branch, as would occur in the absence of selection. Reversions (that is, second mutation events at the same locus in strains already mutated at that locus) were rare and were therefore neglected; this assumption would only serve to increase the apparent independent emergence rate of the simulation. We then used our emergence inference algorithm to analyze the results of the simulation and compared these results with those for the true distributions [Fig. 6]. The observed rate of multiple emergence events was above the neutral expectation throughout the genome [Supplementary Figure 3], ruling out local effects driving the genome-wide observations. Also shown in Supplementary Figure 3 is an example local phylogeny illustrating correct inference of seven alternate alleles and four emergence events for the allele at chromosome X, position 618,548.

**Simulation of admixed populations.** The possibility of admixture between S. cerevisiae strains was simulated as follows. First, five founding lineages were defined, each with 30,000 random polymorphisms generated as described above. Next, in silico meiosis was conducted to generate ten mosaic lineages. The parents of each mosaic lineage were chosen at random from the five founding strains, and the genotype of the resulting meiotic product was determined by choosing one random meiotic crossover on each of the 16 chromosomes. To each of these 15 lineages were then added 10,000 independent additional polymorphisms, again generated as described above. The final genotype matrix was blinded and passed to the inference algorithm, while the true extent of homoplasy was calculated separately (as each true de novo SNP was recorded as it occurred).

**Simulation of balancing selection on ancestral polymorphism.** To simulate the maintenance of ancestral polymorphism due to balancing selection, we first generated simulated admixed populations of five founding and ten admixed lineages as described above. To the $N_{total}$ variants present across the population at this stage, we added $N_{total}f_{ancestral}$ variants with molecular character simulated at random as described above. Balancing selection from the perspective of population sampling was simulated by a random draw for each isolate: at each ancestrally polymorphic locus, each in silico strain was assigned the alternate genotype at random in proportion to $MAF_{balanced}$, reflecting the odds of obtaining an isolate with the given genotype when isolating and sequencing single colonies from a mixed population. The key parameters $f_{ancestral}$ and $MAF_{balanced}$ were varied across a wide range of values, as described in the text, to reflect the current uncertainty regarding their true values in nature.

**Analysis of correlation between ecological niche and genotype.** Ecological niches of each SGRP strain were reported previously[15]. For each variant occurring two or three times within the SGRP collection, the ecotypes of those two or three strains were retrieved, and we determined whether both or all three strains were isolated from the same ecological niche. The neutral expectation in the absence of selection was generated much as described above for the inference of apparent multiple emergence events. Again, 240,000 variants were assumed to occur in ancestors located on branches in the phylogenetic tree with probability proportional to their neighbor-joining branch length. These variants were propagated forward to all strains descended from that branch, simulating the absence of selection. The enrichment of variants occurring two or three times was calculated by the same procedure as for the real genotypes; this was repeated 50 times to generate the random expectations shown in Fig. 7. Variants occurring in four or more isolates were not considered, as only two ecological niches are represented by more than three isolates in the SGRP collection; the lack of representation of other niches confounds the generation of appropriate neutral expectations in these cases.

**Calculation of number of generations required to explore every SNP in the S. cerevisiae genome.** We assume an effective population size of $N_e$ ~$10^6$ individuals[43], each with a genome of $1.2 \times 10^7$ base pairs. Given a mutation rate of $5 \times 10^{-10}$ per base pair per generation[42], each genome experiences ~$6 \times 10^{-3}$ mutations per generation, and the population as a whole undergoes ~$6 \times 10^3$ mutations per generation. Therefore, each base pair of the genome will be mutated in the population within ~$2 \times 10^3$ generations. Accounting for the possibility of

three destination base pairs at each locus yields an estimate of ~$6 \times 10^3$ (~$10^3$–$10^4$) generations.

**Computational methods and resources**. Plate-based growth assays were analyzed using the SGATools image analysis webserver (http://sgatools.ccbr.utoronto.ca/). The forward selection routine used the STEPWISEFIT function in MATLAB; subsequent fine mapping used the ANOVA function. All other analyses were performed and plots generated using custom code in MATLAB (https://github.com/cjakobson/mapping). Most computation was performed locally (Intel Core i7 @ 2.7 GHz; 16 GB RAM; 500 GB local storage); however, some memory-intensive manipulations were performed using Stanford University's Sherlock super-computing cluster (Intel Xeon E5-4640 @ 2.40 GHz (8 core/socket); 64GB RAM; 13 TB local storage).

**Reporting summary**. Further information on experimental design is available in the Nature Research Reporting Summary linked to this article.

## Code availability

All custom code is freely available on GitHub (https://github.com/cjakobson/mapping). The repository also contains instructions for downloading all genotype and phenotype data.

## Data availability

All genotype and phenotype data for the mapping experiments is freely available (see https://github.com/cjakobson/mapping). All other raw data is available in the accompanying Source Data File. Please contact Daniel F. Jarosz (jarosz@stanford.edu) to obtain the $F_6$ segregant panel. All other relevant data is available upon request.

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

## Acknowledgments

The authors gratefully acknowledge José Aguilar-Rodriguez, Rebecca Zabinsky, Zachary Harvey, Alan Itakura, Gavin Sherlock, and Helen Murphy for critical review of the manuscript and members of the Jarosz Laboratory for stimulating discussions. We also thank Joseph Schacherer for assistance in accessing genotype information related to the 1002 Yeast Genomes Project. Some of the computing for this project was performed on the Sherlock cluster. We would like to thank Stanford University and the Stanford Research Computing Center for providing computational resources and support that contributed to these research results. C.M.J. was supported by the National Institutes of Health (NIH-1F32-GM125162 to C.M.J.). R.S. was supported by the Gerald J. Lieberman Fellowship and the Stanford Graduate Fellowship. D.F.J. was supported by the National Institutes of Health (NIH-DP2-GM119140 to D.F.J.), the National Science Foundation (NSF-MCB116762 to D.F.J.), a Searle Scholar Award (14-SSP-210 to D.F.J.), a Kimmel Scholar Award (SFK-15–154 to D.F.J.), and a Science and Engineering Fellowship from the David and Lucile Packard Foundation.

## Author contributions

C.M.J., R.S., and D.F.J. conceived and designed the project. C.M.J. performed the experiments. R.S. designed and implemented the custom code for QTN mapping. C.M.J. designed and implemented the custom code for phylogenetic analysis. C.M.J. and D.F.J. wrote the manuscript. C.M.J., R.S., and D.F.J. revised the manuscript. D.F.J. supervised the project.

## Additional information

**Competing interests:** The authors declare that no competing interests.

