## [Peer Review File · Nature Communications]

Reviewers' comments:

Reviewer #1 (Remarks to the Author):

The manuscript describes widespread homoplasy among yeast strains. QTN were mapped in a large cross and a large fraction were inferred to have originated multiple times using "clean" yeast lineages derived from a neighbor-joining tree. This result implies selection acting on QTN, which is entirely expected based on current quantitative genetic models which either assume mutation selection balance or some form of balancing selection maintain quantitative trait variation in natural populations. What is surprising is that the QTN are inferred to have emerged multiple times, indicating parallel evolution in different lineages. However, as described below the evidence for homoplasy is weak as admixture and recombination are not fully accounted for.

In phylogenetic analysis, homoplasy can be inferred from species trees as species are known to be reproductively isolated. Even so, ancestral polymorphism (present at the time of lineage splitting) is a known confounder that can cause the appearance of homoplasy. The simulations and interpretations of the results do not account for this possibility.

The assumption that "clean" ie non-mosaic (admixed) strains can be treated in a manner similar to species is not correct. The use of these "clean" lineages as well as a chromosome window analysis were used in an attempt to mitigate the possibility that the same variant occurs in different lineages by admixture and recombination rather than parallel evolution from two different mutations. However, the simulations used to justify the analysis did not incorporate admixture, which could be quite common even among clean lineages. Furthermore, both SK1, Y55, W303 were noted to be admixed previously (Liti et al. 2009). W303, SK1 and Y55 are among the strains where multiple emergence is noted in Figure 3 and Figure 4 implying that these may not be parallel changes. Perhaps one of the most disturbing results (indicating incorrect inference of homoplasy) is that not only do QTN show homoplasy but it is rampant across the genome. This observation almost by definition indicates some small amount of admixture between the strains used, not enough to alter the tree, but enough to explain the data.

In regards to the interesting cases of epistasis, independent validation is a gold standard for QTN analysis but was not completed. The single case of validation at UPC2 in the prior publication is great, but the new cases with unexpected effects should be validated; we don't get to prove one QTN and point to it as validation for every subsequent study.

Pg 4 "contrary to expectation that natural variation is dominated by neutral mutations" - while this statement is true for most variation it refers to QTN alleles and it is widely accepted that SNPs that underlie quantitative trait variation are under some form of selection.

Pg 7 "nonlinear epistasis" epistasis by definition is deviation from an additive model, so this wording is redundant or not clear.

Pg 27 "maximum likelihood estimate of the genotype" the method should be described or the software and parameters used to obtain this estimate.

Reviewer #2 (Remarks to the Author):

The submission by Jakobson et al provides some thought-provoking perspectives on natural variation

in a key model species. The main result, that the role of selection in this model may have been greatly underestimated, is of broad interest. I am quite appreciative of the submission, but do have a number of comments and suggestions for improvement. Some of these relate to the key finding of the paper and are critical to address. This may require revising and redoing some of the modelling.

Specific comments, roughly in decreasing order of importance.

1. I am not yet entirely convinced by the conclusion of identical mutations having emerged independently in multiple lineages. To credibly support this rather large claim, the authors need to reject the null hypothesis that mutations emerged once. First, this requires accounting for the fact that mutations may have emerged once in the ancestral lineage but without reaching fixation (i.e. incomplete lineage sorting). Mutations may then segregate as polymorphisms following branching events, potentially giving rise to the seemingly spurious variant distributions observed by the authors. This must be thoroughly rejected. Second, the authors must also reject the possibility that mutations emerged once and then spread across branches through admixture. The fact that some of the investigated populations are referred to as clean do not mean that admixture between them is zero: this labelling was intended to distinguish them from heavily admixed lab and industrial strains, not to claim zero gene flow. Admixture must be explicitly rejected as an explanation for the variant distributions observed - before the claim of homoplasy can be accepted.

The modelling is insufficiently well described and validated. It is highly unclear to what extent it accounts for different degrees and types of incomplete lineage sorting and admixture. The authors need to show that their modelling is done in such a way that both incomplete lineage sorting and admixture can be excluded as alternative explanations for their observations. This will likely require redoing much of the modelling with stricter parameter settings and assumptions.

2. The authors motivate their study with the need to consider genetic variants for traits under selection. Unfortunately, they provide no evidence that the traits considered are under selection, but asks the reader to accept this statement a priori. The authors could quite easily detect trait selection by considering the directionality of QTN's (PMID: 9691061) or effects on growth at different stages (PMID: 25349282). I also wonder if the QTN's detected in their previous paper, for phenotypes they argue should not be under selection, shows the same patterns (Fig 4) as those here reported? This could either strengthen or weaken arguments?

3. The section on epistasis (Fig 2) looks a bit unfinished and is somewhat underwhelming. First, some broader perspectives would be welcome; there must be more than these two "dramatic examples of large effect"? What is the general picture? How common are the phenomena the authors highlight? Right now, it is not clear that the paper adds much to what has already been stated in the refs cited (authors may want to add PMID: 27804950). Second, I lack a statistical perspective on the deviations from additive effects in Fig 2A and 2D – it is not immediately obvious to the reader that these are "dramatic examples of large effect". What is the unit on the y-axis? What do values correspond to in terms of population size? Are they present at more than one time point? What selection coefficient do they correspond to? On what basis are these considered large and dramatic? In general, I feel that the authors are overselling the epistasis aspect of this paper and that it does not contribute much to the storyline. Fig 2C is not sufficiently well described: what data is this based on, and what is the color scale? Is the presumed effect significant – is it immediately clear from the figure that this is a true effect? Neither Fig 2B nor 2C contribute anything of definite value to the storyline and the associated results look and sound circumstantial. There are supplementary figs that contribute more to the storyline. Do Fig 2D really show 2% raffinose, as stated?

4. "To simulate the neutral expectation for the frequency of apparent multiple emergence events within the SGRP collection in the absence of selection, we randomly distributed 240,000 variants across a virtual genome of 12 Mbp." How was this done? Under what assumptions? What were the properties of this virtual genome? How were different types of mutation biases (e.g chromosome region, site, nucleotide type, transition, transversion, neighboring nucleotides) accounted for? The level of detail is clearly insufficient .

5. The M&M for the experimental section is a bit sparse. Genotypes are not stated. Standardization and randomization of replicates and samples is unclear. Was e.g. the medium buffered? The time points for image capture are not specified. It is not stated what growth phase populations are for the images chosen for analysis – do the authors see any problems with comparing environments for which populations are in completely different physiological states at the time of measure? This applies e.g. to Fig 1B and 1C. It is not stated what colony sizes correspond to: cell counts or pixel counts. If the latter, is there a danger in that? It is not clear how well the normalization works in these environments – does it bring down false positive rates to those expected from test cut-offs - i.e. are the authors detecting true biological effects or bias? For instance – the effects of uneven evaporation of ethanol over 120h from different plate sections. Along those lines: can these yeast really grow on lactate? Or is growth on resources carried over from the pre-culture?

6. In the abstract the authors find it important to highlight the number of loci mapped but do not quantify the aspects that directly relate to the main findings of the paper, sticking to "many" and "broadly". I think the author's summary statement in the end of the introduction more clearly and concretely presents their findings. I

7. The first phylogenetic section "Many QTN are not unique to the parent strains" suffers from lack of quantification, instead focusing on examples. What numbers hide behind "some", "many", "multiple" etc? The phenotype data for these examples is mentioned but not shown – it should be. The reader can currently not evaluate them. What data are the phylogenetic trees in Fig 3A and C based on? What is the scale? In Fig 3A Mdm1 is written in lower case. Why are the genotypes at three positions shown in Fig 3B and D? What is the relevance of the genotypes at the non-mentioned loci (green and pink)? Is the AKL1 1911C>T SNP a switch to a rare or common codon: can effects on protein rather than mRNA abundance really be excluded?

8. Fig 1B. It would help understand the fraction and the relevance of variations if the total number of QTNs for each environment was given (could show this as numbers on top of each bar). It would also be nice with a bar for all environments. Fig 1C "metabolic traits" is likely to be a quite confusing label – the authors estimate growth on carbon substrates. 5-FU is usually abbreviated 5-FOA. Are the 5-FOA typical for other drugs tested? Why not show all, or the average of drugs?

I miss a number for the nucleotide distance between the two parents crossed (it is shown in Fig S8, but not stated in the paper) and a distance metric for the individual phylogenies shown. In the Discussion, it would be nice to see the authors discuss the capacities of their QTN calling method with regards to nucleotide distances. What are the practical limitations? How can it be employed to understand phenotype variation between more distant branches in the tree? How robust are extrapolations from tips of the branches to the tree in general?

Detailed Responses to Reviewers' Comments

Reviewer 1

“The manuscript describes widespread homoplasy among yeast strains. QTN were mapped in a large cross and a large fraction were inferred to have originated multiple times using “clean” yeast lineages derived from a neighbor-joining tree. This result implies selection acting on QTN, which is entirely expected based on current quantitative genetic models which either assume mutation selection balance or some form of balancing selection maintain quantitative trait variation in natural populations. What is surprising is that the QTN are inferred to have emerged multiple times, indicating parallel evolution in different lineages. However, as described below the evidence for homoplasy is weak as admixture and recombination are not fully accounted for.

In phylogenetic analysis, homoplasy can be inferred from species trees as species are known to be reproductively isolated. Even so, ancestral polymorphism (present at the time of lineage splitting) is a known confounder that can cause the appearance of homoplasy. The simulations and interpretations of the results do not account for this possibility.”

We thank the reviewer for raising the important alternative hypotheses of incomplete lineage sorting and balancing selection. We agree that this is important to account for and now include a quantitative model that estimates the extent and strength of ancestral polymorphism and balancing selection that would be required to account for the apparent multiple emergence events we observe [see Figure 6].

“The assumption that “clean” ie non-mosaic (admixed) strains can be treated in a manner similar to species is not correct. The use of these “clean” lineages as well as a chromosome window analysis were used in an attempt to mitigate the possibility that the same variant occurs in different lineages by admixture and recombination rather than parallel evolution from two different mutations. However, the simulations used to justify the analysis did not incorporate admixture, which could be quite common even among clean lineages.”

We agree that we cannot assume that the clean lineages are free of admixture. We therefore now only include estimates of multiple emergence made on the basis of the sliding-window local genomic inference method, which determines the phylogeny independently in the neighborhood of each polymorphism. Additionally, to demonstrate that this sliding-window approach is appropriate to account for admixture, we conducted simulations in which we generated complex population phylogenies by random mutagenesis combined with *in silico* mating to generate ‘mosaic’ genotypes. These simulations indicated that we could indeed accurately infer the extent of true underlying homoplasy despite the convolutions due to admixture [Figure 5].

“Furthermore, both SK1, Y55, W303 were noted to be admixed previously (Liti et al. 2009). W303, SK1 and Y55 are among the strains where multiple emergence is noted in Figure 3 and Figure 4 implying that these may not be parallel changes.”

This is an excellent point. We now include both genome-wide and neighborhood-based phylogenetic trees for the example of apparent multiple emergence shown in the text, illustrating that this concern is appropriately accounted for.

“Perhaps one of the most disturbing results (indicating incorrect inference of homoplasy) is that not only do QTN show homoplasy but it is rampant across the genome. This observation almost by definition indicates some small amount of admixture between the strains used, not enough to alter the tree, but enough to explain the data.”

We appreciate the reviewer’s concern, and in the absence of other information might share it. However, the complex simulated admixed populations were correctly assessed by our updated phylogenetic approach. We therefore believe that the widespread apparent multiple emergence is more likely caused by evolutionarily

significant selection occurring on very large numbers of loci across the genome (resulting in either homoplasy or balanced alleles in the population).

"In regards to the interesting cases of epistasis, independent validation is a gold standard for QTN analysis but was not completed. The single case of validation at UPC2 in the prior publication is great, but the new cases with unexpected effects should be validated..."

This is an excellent point. We now include phenotypic validation and functional characterization for two variants associated with the *SUC2* gene.

"Pg 4 "contrary to expectation that natural variation is dominated by neutral mutations" - while this statement is true for most variation it refers to QTN alleles and it is widely accepted that SNPs that underlie quantitative trait variation are under some form of selection."

We agree with the reviewer on this point and have clarified this sentence to indicate that we rather suggest that a larger fraction of segregating natural variants than often thought are in fact QTNs, and are therefore likely subject to selection.

"Pg 7 "nonlinear epistasis" epistasis by definition is deviation from an additive model, so this wording is redundant or not clear."

We thank the reviewer for catching this error and have adjusted the wording accordingly.

"Pg 27 "maximum likelihood estimate of the genotype" the method should be described or the software and parameters used to obtain this estimate."

We have now included a more detailed explanation of this procedure and other computational models and approaches in the Methods section.

Reviewer 2

"The submission by Jakobson et al provides some thought-provoking perspectives on natural variation in a key model species. The main result, that the role of selection in this model may have been greatly underestimated, is of broad interest. I am quite appreciative of the submission, but do have a number of comments and suggestions for improvement. Some of these relate to the key finding of the paper and are critical to address. This may require revising and redoing some of the modelling."

We thank the reviewer for their enthusiasm about the work, and have endeavored to address each of their comments below.

"1. I am not yet entirely convinced by the conclusion of identical mutations having emerged independently in multiple lineages. To credibly support this rather large claim, the authors need to reject the null hypothesis that mutations emerged once. First, this requires accounting for the fact that mutations may have emerged once in the ancestral lineage but without reaching fixation (i.e. incomplete lineage sorting). Mutations may then segregate as polymorphisms following branching events, potentially giving rise to the seemingly spurious variant distributions observed by the authors. This must be thoroughly rejected. Second, the authors must also reject the possibility that mutations emerged once and then spread across branches through admixture. The fact that some

of the investigated populations are referred to as clean do not mean that admixture between them is zero: this labelling was intended to distinguish them from heavily admixed lab and industrial strains, not to claim zero gene flow. Admixture must be explicitly rejected as an explanation for the variant distributions observed - before the claim of homoplasy can be accepted."

Reproduced from our response to the similar concerns raised by Reviewer 1 above:

We thank the reviewer for raising this critical point. We agree that we cannot assume that the clean lineages are free of admixture. We therefore now only include estimates of multiple emergences made on the basis of the sliding-window local genomic inference method, which determines the phylogeny independently in the neighborhood of each polymorphism. Additionally, to demonstrate that this sliding-window approach is appropriate to account for admixture, we conducted extensive simulations in which we generated complex population phylogenies by random mutagenesis combined with *in silico* mating to generate 'mosaic' genotypes. These simulations indicated that we could indeed accurately infer the extent of true underlying homoplasy despite the convolutions due to admixture [now provided in Figure 5].

"The modelling is insufficiently well described and validated. It is highly unclear to what extent it accounts for different degrees and types of incomplete lineage sorting and admixture. The authors need to show that their modelling is done in such a way that both incomplete lineage sorting and admixture can be excluded as alternative explanations for their observations. This will likely require redoing much of the modelling with stricter parameter settings and assumptions."

Reproduced from our response to the similar concerns raised by Reviewer 1 above:

We thank both reviewers for raising the important alternative hypotheses of incomplete lineage sorting and balancing selection. We now include a quantitative model (thoroughly described in the Methods section) that estimates the extent and strength of ancestral polymorphism and balancing selection that would be required to account for the apparent multiple emergence events we observe [now provided in Figure 6].

"2. The authors motivate their study with the need to consider genetic variants for traits under selection. Unfortunately, they provide no evidence that the traits considered are under selection, but asks the reader to accept this statement a priori. The authors could quite easily detect trait selection by considering the directionality of QTN's (PMID: 9691061) or effects on growth at different stages (PMID: 25349282). I also wonder if the QTN's detected in their previous paper, for phenotypes they argue should not be under selection, shows the same patterns (Fig 4) as those here reported? This could either strengthen or weaken arguments?"

We thank the reviewer for this excellent suggestion. Indeed, we find that a directional test for selection reveals strong coherence in nearby QTLs for metabolic traits, suggesting positive lineage selection on these characteristics [see Figure 3]. The same was not true of the drug and other stressor traits mapped previously, supporting our assertion that the traits examined here are more ecologically relevant.

"3. The section on epistasis (Fig 2) looks a bit unfinished and is somewhat underwhelming. First, some broader perspectives would be welcome; there must be more than these two "dramatic examples of large effect"? What is the general picture? How common are the phenomena the authors highlight? Right now, it is not clear that the paper adds much to what has already been stated in the refs cited (authors may want to add PMID: 27804950). Second, I lack a statistical perspective on the deviations from additive effects in Fig 2A and 2D – it is not immediately obvious to the reader that these are "dramatic examples of large effect". What is the unit on the y-axis? What do values correspond to in terms of population size? Are they present at more than one time point? What selection coefficient do they correspond to? On what basis are these considered large and dramatic? In general, I feel that the authors are overselling the epistasis aspect of this paper and that it does not contribute much to the storyline. Fig 2C is not sufficiently well described: what data is this based on, and what is the color scale? Is the presumed effect significant – is it immediately clear from the figure that this is a true effect? Neither

Fig 2B nor 2C contribute anything of definite value to the storyline and the associated results look and sound circumstantial. There are supplementary figs that contribute more to the storyline. Do Fig 2D really show 2% raffinose, as stated?"

We appreciate the reviewer's concern, and now provide a stronger example with molecular validation (*SUC2*) in Figure 2, simultaneously addressing the formatting issues raised here. We would be happy to move some of the supplementary figures into the main text at the discretion of the editor.

We did not conduct a genome-wide test for epistasis between QTNs because we are underpowered to do so, owing to the large number of tests required because of the low linkage in our segregant panel. Instead we inspected our QTN data to find examples of QTNs affecting the same gene, which we reasoned might be likely to exhibit epistasis. We suspect that evidence of epistasis between even the few neighboring QTNs we examined suggests that not only are compound QTLs containing multiple QTNs common (Sharon, *et al Cell* 2018), but that these loci may often contain substantial nonlinearities to which lower-resolution approaches are blind.

It is challenging to convert the relative growth differences in terms of relative colony size, as shown for instance in Figures 1 and 2, to selection coefficients, except to say that the two are likely monotonically related. In future studies we hope to compare the regression coefficients of mapped QTNs to selection coefficients from genome editing experiment in high-throughput to more thoroughly define this correlation.

"4. "To simulate the neutral expectation for the frequency of apparent multiple emergence events within the SGRP collection in the absence of selection, we randomly distributed 240,000 variants across a virtual genome of 12 Mbp." How was this done? Under what assumptions? What were the properties of this virtual genome? How were different types of mutation biases (e.g chromosome region, site, nucleotide type, transition, transversion, neighboring nucleotides) accounted for? The level of detail is clearly insufficient."

We sincerely apologize for this oversight. Significant additional details on the generation of the random mutations, specifying these parameters, are now included in the Methods.

"5. The M&M for the experimental section is a bit sparse. Genotypes are not stated. Standardization and randomization of replicates and samples is unclear. Was e.g. the medium buffered? The time points for image capture are not specified. It is not stated what growth phase populations are for the images chosen for analysis – do the authors see any problems with comparing environments for which populations are in completely different physiological states at the time of measure? This applies e.g. to Fig 1B and 1C."

We thank the reviewer for raising this important concern. We have expanded this section in the Methods, and now include growth data and mapping performed in each environment at 24-hour intervals.

"It is not stated what colony sizes correspond to: cell counts or pixel counts. If the latter, is there a danger in that? It is not clear how well the normalization works in these environments – does it bring down false positive rates to those expected from test cut-offs - i.e. are the authors detecting true biological effects or bias? For instance – the effects of uneven evaporation of ethanol over 120h from different plate sections."

The colony sizes are in pixels; we think it likely that the corresponding measurements are monotonically, but not entirely linearly, related to cell number. Empirically, we and others have observed that normalized phenotypes correspond well between growth in liquid vs. on solid medium (see VanderSluis *et al. Genome Biology* 2014; She and Jarosz *Cell* 2018), suggesting that this effect is not severely confounding.

Our regression includes 'pseudo-genotypes' corresponding to plate-to-plate variability (possibly attributable to position within the incubator or variation in inoculum size) and plate-edge effects (possibly due to uneven evaporation as the reviewer mentions or variation in oxygen availability in the plate center). These 'pseudo-

alleles' do sometimes explain a small portion of the phenotypic variance, suggesting that these effects are both significant and accounted for by our procedure.

"Along those lines: can these yeast really grow on lactate? Or is growth on resources carried over from the pre-culture?"

We thank the reviewer for catching this oversight. Having now compared the growth of the segregant panel on lactic acid and on media lacking a carbon source entirely, we believe the reviewer to be correct that the yeast do not substantially metabolize the lactic acid. For this reason, we have substituted a new carbon source, 2% sucrose, in the phenotyping experiment used to generate the time-resolved mapping data included in this revision. The role of pH in growth on varying carbon sources remains an interesting area of exploration for future experiments.

"6. In the abstract the authors find it important to highlight the number of loci mapped but do not quantify the aspects that directly relate to the main findings of the paper, sticking to "many" and "broadly". I think the author's summary statement in the end of the introduction more clearly and concretely presents their findings...."

We now include these precise figures in the summary as the reviewer suggests.

"7. The first phylogenetic section "Many QTN are not unique to the parent strains" suffers from lack of quantification, instead focusing on examples. What numbers hide behind "some", "many", "multiple" etc? The phenotype data for these examples is mentioned but not shown – it should be. The reader can currently not evaluate them."

We have updated Figure 1 to highlight the phenotypic data for various example QTNs of diverse molecular mechanisms in a selection of growth conditions and time points. We have also added specific numbers to this section as suggested by the reviewer.

"What data are the phylogenetic trees in Fig 3A and C based on? What is the scale? In Fig 3A Mdm1 is written in lower case. Why are the genotypes at three positions shown in Fig 3B and D? What is the relevance of the genotypes at the non-mentioned loci (green and pink)? Is the AKL1 1911C>T SNP a switch to a rare or common codon: can effects on protein rather than mRNA abundance really be excluded?"

We now show phylogenetic trees based on both the genome-wide and sliding-window genotype matrices for the *IMA1* polymorphism of interest, illustrating that the polymorphism in question are indeed the result of apparent homoplasy after accounting for admixture.

We agree that other molecular mechanisms are possible; we have modified the text to elaborate on this possibility.

"8. Fig 1B. It would help understand the fraction and the relevance of variations if the total number of QTNs for each environment was given (could show this as numbers on top of each bar). It would also be nice with a bar for all environments."

We thank the reviewer for this suggestion. The total number of identified QTNs is now included in the figure for all environments, as is a summary bar showing the spectrum across all identified QTNs.

"Fig 1C "metabolic traits" is likely to be a quite confusing label – the authors estimate growth on carbon substrates. 5-FU is usually abbreviated 5-FOA. Are the 5-FOA typical for other drugs tested? Why not show all, or the average of drugs?"

We thank the reviewer for this important point. In this case, the drug in question is 5-fluoro-uracil (5-FU), as distinct from 5-fluoro-orotic acid (5-FOA). Per the reviewer's suggestion, we now include the variance scree for each condition tested in She and Jarosz as well as for each condition tested here.

"I miss a number for the nucleotide distance between the two parents crossed (it is shown in Fig S8, but not stated in the paper) and a distance metric for the individual phylogenies shown."

We apologize for this oversight. The two parents differ by 12,054 polymorphisms; this information is now included and nucleotide distance scale bars are now included in each phylogeny shown.

"In the Discussion, it would be nice to see the authors discuss the capacities of their QTN calling method with regards to nucleotide distances. What are the practical limitations? How can it be employed to understand phenotype variation between more distant branches in the tree? How robust are extrapolations from tips of the branches to the tree in general?"

We thank the reviewer for this excellent suggestion, and now elaborate on these points in the Discussion. Briefly, increasing numbers of polymorphisms will necessitate both more haploid progeny (to maintain sensitivity to variants of small effect) and more rounds of inbreeding (to disrupt linkage sufficiently maintain similar QTN-resolving power). To understand more distant branches, similarly sized crosses of pairs of neighboring strains from divergent branches could be constructed, or very large panels could be generated to span the large genetic distances between clades. The extent to which molecular spectra of QTNs are maintained across clades, and between ancestral and derived alleles that are QTNs, are key questions for future research, and an active area of investigation in our laboratory.

REVIEWERS' COMMENTS:

Reviewer #2 (Remarks to the Author):

The revised submission by Jakobson et al is a much improved version of the already interesting, but not sufficiently well supported by, original version. The authors have gone to substantial lengths to address my concerns. Doing so, they have added extensive additional empirical and analytical support for their key findings, replaced incorrect data, and much improved the text in various sections.

I note that important aspects of their argumentation relies on modeling evolution, which always relies on assumptions, simplifications and approximations of parameters for which empirical support often is lacking. Such argumentation therefore always comes with an element of doubt attached. In this case, I find it slightly disconcerting that *S. paradoxus* SNPs follow a very different pattern of distribution. This may reflect differences in degrees of ecological differentiation, as suggested by the authors. An alternative explanation is that admixture is not fully accounted for and that the very different levels of admixture in the two species explains the dramatic differences observed.

Nevertheless, I do believe that the authors have done all that could reasonably be asked of them, and all that I have asked of them. I am now convinced that the yeast phenotypes studied have been under selection. I am also convinced that the QTNs controlling variation in these phenotypes have been under selection. Finally, I find the reasons to believe that a surprisingly large fraction of yeast SNPs, in general, have been under selection, to be sufficiently compelling for publication.

This paper is an important contribution to the debate of the relative roles of selection and drift in controlling variation and should be published by Nat Com.

Minor issues that should not delay publication:

Page 7 row 116. I get a bit confused by the statement(s) on which allele is rare/common. The confusion may be mine, but I ask the authors to ensure that there is no contradiction between text and figures.

Fig 2(A). I note that the arrangement of replicates shown implies that the four replicates i) may have been derived from the same pre-culture and ii) are arranged in the same plate region on the same plate. In my view this gives a misleading and overly optimistic view of variation between replicates relative between different samples. A thoroughly randomized design, with pre-culture replication, would have been preferable. While the findings presented probably are robust to this slight oversight, I am not convinced that it is correct to present the design as true biological replication.

Response to reviewer comments

REVIEWERS' COMMENTS:

Reviewer #2 (Remarks to the Author):

The revised submission by Jakobson et al is a much improved version of the already interesting, but not sufficiently well supported by, original version. The authors have gone to substantial lengths to address my concerns. Doing so, they have added extensive additional empirical and analytical support for their key findings, replaced incorrect data, and much improved the text in various sections.

We thank the reviewers again for their insightful comments, which were instrumental in improving the manuscript.

*I note that important aspects of their argumentation relies on modeling evolution, which always relies on assumptions, simplifications and approximations of parameters for which empirical support often is lacking. Such argumentation therefore always comes with an element of doubt attached. In this case, I find it slightly disconcerting that *S. paradoxus* SNPs follow a very different pattern of distribution. This may reflect differences in degrees of ecological differentiation, as suggested by the authors. An alternative explanation is that admixture is not fully accounted for and that the very different levels of admixture in the two species explains the dramatic differences observed.*

We agree that we can never be sure that we have fully accounted for the complexities undoubtedly present in the wild. We now comment on this concern in the Discussion.

Nevertheless, I do believe that the authors have done all that could reasonably be asked of them, and all that I have asked of them. I am now convinced that the yeast phenotypes studied have been under selection. I am also convinced that the QTNs controlling variation in these phenotypes have been under selection. Finally, I find the reasons to believe that a surprisingly large fraction of yeast SNPs, in general, have been under selection, to be sufficiently compelling for publication.

This paper is an important contribution to the debate of the relative roles of selection and drift in controlling variation and should be published by Nat Com.

Minor issues that should not delay publication:

Page 7 row 116. I get a bit confused by the statement(s) on which allele is rare/common. The confusion may be mine, but I ask the authors to ensure that there is no contradiction between text and figures.

We have double-checked this point and found that the *SUC2*⁻⁶ alleles in Fig. 2B were mis-colored. This has been corrected.

Fig 2(A). I note that the arrangement of replicates shown implies that the four replicates i) may have been derived from the same pre-culture and ii) are arranged in the same plate region on the same plate. In my view this gives a misleading and overly optimistic view of variation between replicates relative between different samples. A thoroughly randomized design, with

pre-culture replication, would have been preferable. While the findings presented probably are robust to this slight oversight, I am not convinced that it is correct to present the design as true biological replication.

We have corrected the figure legend to indicate that the quadruplicates shown are indeed technical replicates, as the reviewer points out, but are representative of the N = 8 biological replicates performed.